# Five-to-Fifteen—Parental Perception of Developmental Profile from Age 5 to 8 Years in Children Born Very Preterm

**DOI:** 10.3390/jpm13050819

**Published:** 2023-05-12

**Authors:** Eeva Mäkilä, Mikael O. Ekblad, Päivi Rautava, Helena Lapinleimu, Sirkku Setänen

**Affiliations:** 1Department of Pediatric Neurology, University of Turku, 20014 Turku, Finland; 2Salo Health Centre, 24240 Salo, Finland; 3Department of General Practice, Institute of Clinical Medicine, Turku University Hospital, University of Turku, 20014 Turku, Finland; 4Public Health, Turku Clinical Research Centre, Turku University Hospital, University of Turku, 20014 Turku, Finland; 5Department of Pediatrics, Turku University Hospital, University of Turku, 20014 Turku, Finland

**Keywords:** developmental assessment, FTF, parental questionnaire, prospective, very low birth weight (VLBW)

## Abstract

Children born very preterm have increased risk of developmental difficulties. We examined the parental perception of developmental profile of children born very preterm at 5 and 8 years by using the parental questionnaire Five-to-Fifteen (FTF) compared to full-term controls. We also studied the correlation between these age points. The study included 168 and 164 children born very preterm (gestational age < 32 weeks and/or birth weight ≤ 1500 g) and 151 and 131 full-term controls. The rate ratios (RR) were adjusted for sex and the father’s educational level. At 5 and 8 years, children born very preterm were more likely to have higher scores (more difficulties) compared to controls in motor skills (RR = 2.3, CI 95% = 1.8–3.0 at 5 years and RR = 2.2, CI 95% = 1.7–2.9 at 8 years), executive function (1.7, 1.3–2.2 and 1.5, 1.2–2.0), perception (1.9, 1.4–2.5 and 1.9, 1.5–2.5), language (1.5, 1.1–1.9 and 2.2, 1.7–2.9), and social skills (1.4, 1.1–1.8 and 2.1, 1.6–2.7), and at 8 years in learning (1.9, 1.4–2.6) and memory (1.5, 1.2–2.0). There were moderate-to-strong correlations (r = 0.56–0.76, *p* < 0.001) in all domains between 5 and 8 years in children born very preterm. Our findings suggest that FTF might help to earlier identify children at the greatest risk of incurring developmental difficulties persisting to school-age.

## 1. Introduction

Children born very preterm have increased risk of neurodevelopmental impairments [1,2] including cerebral palsy and severe cognitive impairment. Despite the decreasing prevalence of cerebral palsy (CP) [3,4], children born very preterm are still more often susceptible to difficulties, for example, in motor and language skills, cognition, and attention, compared to children born full-term [5,6,7]. Such difficulties may become more prominent with increasing demands at school-age and have a negative impact on daily activities and education [8,9,10,11]. Previous studies have reported that developmental difficulties continue into adulthood [12,13] and should therefore be recognized and supported as early as possible.

Previous studies have shown that parental questionnaires can be an easy, reliable, and often low-cost tool when assessing the development of 2–15-year-old children born preterm [14,15]. The Five-to-Fifteen (FTF) questionnaire is validated for identifying developmental problems in children between 5 and 17 years [16,17,18]. FTF scores have been shown to correlate with the corresponding scores of the neuropsychological assessment instrument NEPSY in 5-year-old children [16]. Most of the previous literature regarding the FTF questionnaire concerning very or extremely preterm children have only studied selected domains of the FTF [19,20,21]. Rautava et al. [22] reported on all the FTF domains and their results showed more difficulties in 5-year-old children born very preterm compared to full-term controls. We could not find previous prospective studies comparing the whole FTF profile of children born very preterm and controls at different age points nor with the aim of discovering correlations between the FTF scores at different age points. Information about parental perception of developmental difficulties and their persistence into school age might enable early identification of developmental problems in order to allocate resources for timely support in children born very preterm.

Our aim was to describe the FTF profile in children born very preterm at 5 and 8 years compared to full-term controls. The hypothesis was that children born very preterm would have more difficulties across the whole FTF profile compared to the controls at both ages. We also aimed to study the correlations of FTF results between 5 and 8 years. We hypothesized a strong correlation in FTF results between these two age points.

## 2. Materials and Methods

### 2.1. Study Design

In this study, the parents of 168 children born very preterm and 151 full-term controls completed the FTF questionnaire at 5 years, and 164 and 131 at 8 years, respectively. We studied the difference between these groups, and the correlations between the age points. All children whose parent(s) had completed the FTF at 5 and/or at 8 years were included. No power analysis was performed.

This study is part of the multidisciplinary longitudinal PIPARI project (development and functioning of very low birth weight infants from infancy to school age) including very preterm infants born at Turku University Hospital, Finland, between 2001 and 2006. The Ethics Review Committee of the Hospital District of Southwest Finland approved the study protocol in 2000. Written informed consent for this study was obtained from all the parents.

### 2.2. Participants

The study sample consisted of 289 very preterm infants (gestational age < 32 weeks and/or birth weight ≤ 1500 g) born in Turku University Hospital, Finland, between 2001 and 2006. At the time when the parents returned the questionnaires, the mean (SD), [min, max] age of the children born very preterm was 5.0, (0.1), [4.9–5.8] and 7.8, (0.3), [7.2–8.6] years old. A flowchart of the studied children is shown in Figure 1.

The control group consisted of 200 infants (100 male; 100 female) born between 2001 and 2003 at Turku University Hospital. The comparison group was recruited by asking the parents of the first male and the first female born each Monday to take part in the study. If they refused, the parents of the next male or female were asked. Full-term controls were at born ≥ 37 weeks of gestational age into Finnish and/or Swedish-speaking families and were not admitted to a neonatal intensity care unit during their first week of life. The exclusion criteria were congenital anomalies or syndromes, the mother’s self-reported use of illegal drugs or alcohol during pregnancy, and a birth weight < −2.0 SD (small for gestational age) according to age- and gender-specific Finnish growth charts. The flowchart of the full-term controls is shown in Figure 2.

### 2.3. Evaluations

The FTF questionnaire is a validated instrument that was developed by a group of researchers form the Nordic countries of Sweden, Finland, Denmark, and Norway [16,17,18]. The original version of the questionnaire was constructed and standardized in 2004, and a revised version was upgraded and re-standardized in 2016 [23]. It contains 181 questions about a child’s development, designed to be completed by parents or teachers observing the child daily. The questions are answered according to the child’s strengths and weaknesses during the previous six months. There are three alternative answers for each statement in the questionnaire: not true/never (0 points), somewhat true/sometimes (1 point), and very true/often (2 points). More points indicate more difficulties.

The questionnaire includes 8 domains (and subdomains): (1) motor skills (gross motor skills and fine motor skills), 17 statements; (2) executive functions (attention and concentration, overactivity and impulsivity, passivity and inactivity, and planning and organizing), 25 statements; (3) perception (perception of space and direction, concepts of time, perception of own body, and perception of visual forms and figures), 18 statements; (4) memory, 11 statements; (5) language (comprehension of spoken language, expressive language, verbal communication), 21 statements; (6) learning (reading and writing, arithmetic, general learning, and coping with learning), 27 statements; (7) social skills, 27 statements; (8) emotional/behavioral difficulties (internalization, acting out, and obsessive actions or thoughts), 33 statements. The domain of learning and the subdomain of concepts of time were omitted from the analyses of 5-year-old’s questionnaires as recommended by the manual [23].

### 2.4. Data Analysis

The statistical analyses were performed using SPSS version 28.0 (IBM SPSS Statistics, IBM Corporation, Armonk, NY, USA). Two-tailed *p*-values of <0.05 were considered statistically significant.

The difference in continuous background characteristics between the children born very preterm participating in the study and those who withdrew were analyzed using either the two-sample *t*-test when variables were normally distributed or the Mann–Whitney U Test when variables were not normally distributed. For the categorical background characteristics, a chi-square test or Fisher’s exact test was used. The distribution of continuous variables was studied both graphically and using the Shapiro–Wilk test.

Generalized linear models were used to study the effect of very preterm birth on the FTF profile at both age points. The response distribution of the totals of the FTF scores was negative binomial, and the link function was logarithmic. In the analysis of the FTF scores, the logarithm of the number of answered questions was used as the offset variable. The results of these comparisons are provided as rate ratios (RR) with 95% confidence intervals (CI) and *p*-values. If more than 40% of the answers in any FTF domain were missing, the child concerned was not included in the analyses of that domain, as suggested by the instrument’s developers. The analyses were adjusted for sex and the father’s educational level; this was because male sex and the father’s low educational level have been shown to affect many domains of FTF [17].

Correlations between the FTF scores at 5 and 8 years were calculated using Pearson’s rho. Only the children whose parents completed the questionnaire at both age points were included when the correlations were calculated.

## 3. Results

A total of 168 and 164 children born very preterm were included at 5 and 8 years, respectively. The background characteristics of these children and controls are shown in Table 1. The included children at 5 years were more often born to mothers who had received prenatal corticosteroids (95.8% vs. 86.8%, *p* = 0.002), children who were born with lower birth weight (1104.7 g vs. 1260.1 g, *p* = 0.003), had sepsis more often (20.8% vs. 3.8%, *p* = 0.003), a higher full-scale intelligence quotient (mean 102.1 vs. 95.1, *p* = 0.04), and had CP less often (4.8% vs. 13.7%, *p* = 0.03) than children who withdrew. The mothers’ educational level higher was more often in the study children than those who withdrew both at 5 years (48.1% vs. 19.1%, *p* < 0.001) and 8 years (46.5% vs. 26.0%, *p* = 0.01).

### 3.1. Five-to-Fifteen at 5 Years

The mean scores of the FTF in the children born very preterm and the full-term controls at 5 years are shown in Table 2. The children born very preterm had higher mean scores (more difficulties) compared to the controls in all domains and subdomains of the FTF in motor skills, executive functions, perception, memory, language, social skills and emotional/behavioral difficulties. The most affected domains in children born very preterm (FTF-adjusted mean scores that were more than two times higher compared to the controls) were motor skills (RR = 2.3, 95% CI = 1.8–3.0) including gross motor skills (2.5, 1.9–3.4), passivity and inactivity (2.3, 1.5–3.4), and perception of visual forms and figures (2.7, 1.8–4.1). The difference between the groups was not statistically significant in memory (*p* = 0.08), emotional/behavioral difficulties (*p* = 0.2), and acting out (*p* = 0.4). When the children with neurodevelopmental impairment were excluded, the difference between the groups was no longer statistically significant in expressive language (*p* = 0.1) and social skills (*p* = 0.1) (Table 2). Figure 3 demonstrates the difference between the levels of FTF scores in children born very preterm and controls at 5 years of age.

### 3.2. Five-to-Fifteen at 8 Years

The mean scores of the FTF in the children born very preterm and the full-term controls at 8 years are shown in Table 3. The children born very preterm had higher mean scores (more difficulties) compared to the controls in all domains and subdomains of the FTF in motor skills, executive functions, perception, memory, learning, language, social skills, and emotional/behavioral difficulties. The most affected domains in children born very preterm (FTF-adjusted mean scores that were more than two times higher compared to the controls) were motor skills (RR = 2.2, 95% CI = 1.7–2.9), gross motor skills (2.7, 2.0–3.8), perception of space and direction (2.7, 1.8–4.0), perception of visual forms and figures (4.1, 2.3–7.3), arithmetic (2.0, 1.4–2.8), general learning (2.3, 1.6–3.4), language (2.2, 1.7–2.9), comprehension of spoken language (2.7, 1.8–3.9), expressive language (2.2, 1.6–3.0), and social skills (2.1, 1.6–2.7). The difference between the groups was not statistically significant in acting out (*p* = 0.2) and obsessive actions and thoughts (*p* = 0.08). When the children with neurodevelopmental impairment were excluded, the difference between the groups was no longer statistically significant in reading and writing (*p* = 0.06) and emotional/behavioral difficulties (*p* = 0.05) (Table 3). Figure 4 demonstrates the difference between the levels of FTF scores of children born very preterm and controls at 8 years of age.

### 3.3. Correlation between Age Points

Correlations between the FTF scores at 5 and 8 years varied (r = 0.39–0.77; *p* < 0.001) in the children born very preterm and (r = 0.19–0.72; *p* = 0.04–<0.001) in the controls as shown in Table 4. The correlation was stronger among the children born very preterm than the controls in every domain and subdomain except in perception of space and direction, memory, verbal communication, emotional/behavioral difficulties, and internalization. The correlation between the FTF scores at 5 and 8 years in the children born very preterm was strong or very strong (r ≥ 0.6) in motor skills (both gross motor skills and fine motor skills), executive functions, memory, language, expressive language, social skills, emotional/behavioral difficulties, and acting out. When the children with neurodevelopmental impairment were excluded, correlations between the FTF scores at 5 and 8 years varied (r = 0.30–0.80; *p* < 0.001) in the children born very preterm.

## 4. Discussion

This study described, for the first time, the entire parental perception of developmental profile of children born very preterm assessed using the parental FTF questionnaire at 5 and 8 years. The children born very preterm had more difficulties in all the FTF domains compared to the controls at both age points, as hypothesized. In addition, significant positive correlations between the FTF scores at 5 and 8 years were found in children born very preterm, also when children with neurodevelopmental impairment were excluded.

At 5 years, the area of Motor skills was the developmental domain most affected in the children born very preterm compared to the full-term controls, especially regarding gross motor skills, where children born very preterm had scores that were more than two times higher, i.e., more difficulties, in comparison with their full-term peers. This finding is in line with previous studies that have reported very preterm children as being approximately six to eight times more likely to have motor impairment or developmental coordination disorder (DCD) than full term controls [7,20,27,28,29]. DCD, in turn, has been reported to associate with a lower full-scale intelligence quotient, a slower procession speed, problems with attention, perception, executive functions, and visuomotor coordination, as well as emotional, social, and behavioral problems in children born very or extremely preterm [10,20]. We have previously shown in this same very preterm PIPARI study cohort that 11-year-old children with DCD had 15 points lower full-scale intelligence quotient and lower self-experienced health-related quality of life [28], as well as more problems with social competence [30] compared to children born very preterm without DCD at 11 years.

Children born very preterm have been reported to have more difficulties in every domain of FTF compared to full term controls at 5 years of age [22]. Our study adds to the previous literature by still showing persisting difficulties at 8 years in all domains in the children born very preterm compared to the controls. At 8 years, language was the most affected developmental domain (together with motor skills) in the children born very preterm compared to the full-term controls, especially regarding comprehension of spoken language (and gross motor skills); in this domain, the children born very preterm had a score that was almost three times higher, i.e., they had more difficulties, in comparison with their full-term controls. This finding is in line with previous studies that report children born very preterm as having weaker language development at 2 years and weaker literacy skills at 7 years compared to the full-term controls [31].

It has been previously reported that even mild-to-moderate difficulties in executive functions affected learning skills in 10–15-year-old children born very preterm compared to full term controls [32]. In the present study, children born very preterm had more difficulties in all subdomains of executive functions (attention and concentration, overactivity and impulsivity, passivity and inactivity, and planning and organizing) at 5 and 8 years compared to the controls. At the age of 8 years, they also had more difficulties in all subdomains of learning that were not included in the questionnaire for 5-year-old children. More research is needed to evaluate whether the FTF questionnaire could be a valuable tool in early identification of later developmental difficulties such as motor impairment or learning difficulties in order for timely support to be provided for the children with the greatest risk. This is also important from the perspective of prevention, as poor school management and lower education levels have great social and economic effects on both the individual and society as a whole.

In this study, children born very preterm had more difficulties in perception at both age points, and in memory at 8 years compared to full-term controls. Previous studies have shown children born very preterm or with very low birth weight to have more difficulties with verbal and visual working memory tasks, especially in cognitively demanding situations [33,34]. Moreover, children born preterm are known to have more social difficulties compared to full-term controls, and these difficulties seem to persist from early childhood into school-age and adulthood [35,36,37,38]. Similarly, the present study showed that children born very preterm had higher scores in social skills compared to controls at both age points. However, the difference did not remain statistically significant at 5 years when children with neurodevelopmental impairment were excluded. In emotional/behavioral difficulties the difference between children born very preterm and full-term controls was statistically significant only in internalization at both age points and obsessive actions and thoughts at the age of 5 years. In contrast, some previous studies have shown children born preterm to have more behavioral and emotional difficulties, especially in terms of hyperactivity, inattention, and difficulties in peer relations, compared to full-term controls [38,39,40,41].

At 5 years, there were no statistically significant differences in domains of memory and emotional/behavioral difficulties between children born very preterm and controls, whereas at 8 years, these differences were statistically significant. This might be due to increasing demands at school-age. In addition, some parts of the children’s development might be more difficult for parents to estimate than others. For example, 20.7% of parents did not recognize the memory difficulties of their children born with extremely low birth weight according to the FTF questionnaire despite abnormal results in the memory domain of the neuropsychological assessment NEPSY-II [42].

We found strong positive correlations between all eight main FTF domains at 5 and 8 years of age in children born very preterm, except in the domain of perception, where the correlation was moderate. The correlations remained statistically significant even when the children with neurodevelopmental impairment were excluded. Correlations were stronger in most FTF domains and subdomains in the very preterm children compared to the full-term controls. The strongest correlations were found in motor skills, language, and social skills in the children born very preterm, and in motor skills and emotional/behavioral difficulties in the controls. Because the difficulties in many developmental domains at the age of 5 years seem to continue into school age, it is important to identify these children as early as possible and provide them support also to prevent later comorbidities.

A major strength of this study was its prospective design and the use of the same validated questionnaire at both age points. In addition, the follow-up rate was reasonably high. If more than 40% of the answers in any FTF domain were missing, the child concerned was not included in the analyses of that domain, which secured the validity of the data. We were also concerned with the number of the answered questions in the analyses, which strengthened the reliability of the results. At both age points, the mothers of the children included in the study were more liable to have a high educational level compared to the mothers of the children who withdrew, which may have had a positive impact on the children’s developmental profile. At 5 years, the studied children had more often been prescribed prenatal corticosteroids, which have been shown to prevent morbidity [43]. In addition, the studied children included had a higher full-scale intelligence quotient and suffered less often from CP than the children who withdrew. However, the studied children had lower birth weights and a higher rate of neonatal sepsis compared to children who withdrew, indicating more risk factors for developmental problems. These differences between the included studied children born very preterm and those who withdrew might affect the generalizability of our results. Another possible limitation of our study was the lack of information as to whether the same parent completed the FTF questionnaire at both age points. Having the same parent to answer the questionnaire at both age points would have strengthened the results.

In this study, we did not aim to compare the FTF scores with neurodevelopmental assessment methods. However, previous studies have reported that parents of extremely preterm children underestimated their children’s cognitive and motor problems [20,42]. In our study, the children born very preterm had higher scores in all FTF domains compared to the controls. A possible parental underestimation could have decreased this difference. However, our full-term controls have been reported to have higher full-scale intelligence quotient compared to general population at 5 years [26], which might have increased the difference in FTF scores in comparison with children born very preterm.

## 5. Conclusions

The FTF questionnaire showed more difficulties according to parental perception throughout the whole 5-year developmental profile in the children born very preterm compared to the full-term controls. These difficulties seemed to persist into school age as the differences were still found at 8 years with a strong correlation between the age points. The FTF might provide a useful tool for the early identification of developmental problems and to allocate resources for timely support and prevention. Our study highlights the importance of identifying neurodevelopmental difficulties in preterm infants at an early stage. Moreover, children with identified difficulties need to be actively referred to rehabilitation.

## Figures and Tables

**Figure 1 jpm-13-00819-f001:**
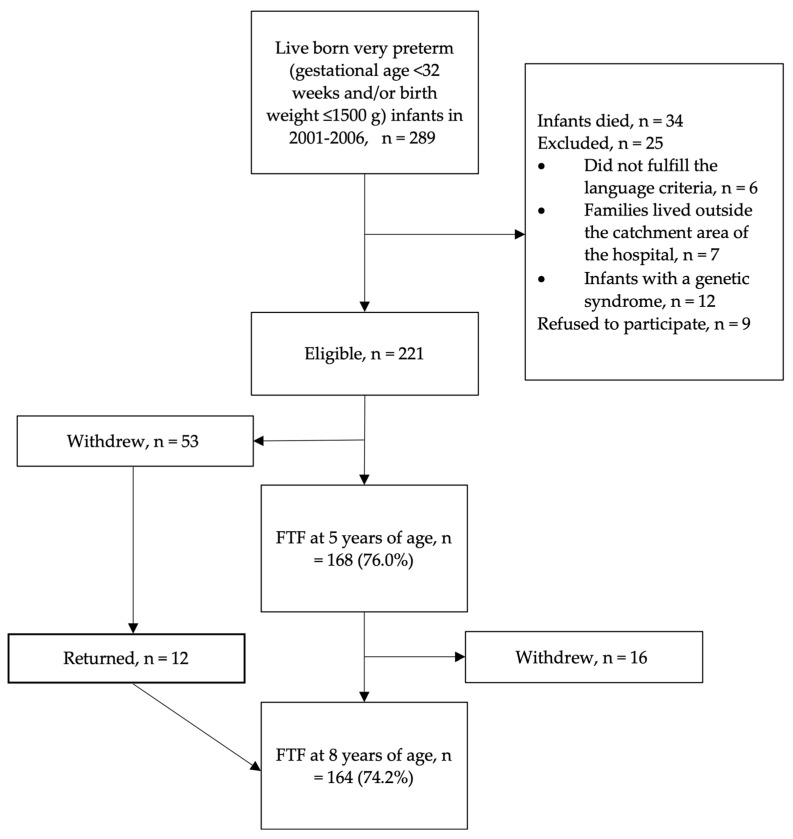
Flowchart of the study children born very preterm. FTF: Five-to-Fifteen questionnaire.

**Figure 2 jpm-13-00819-f002:**
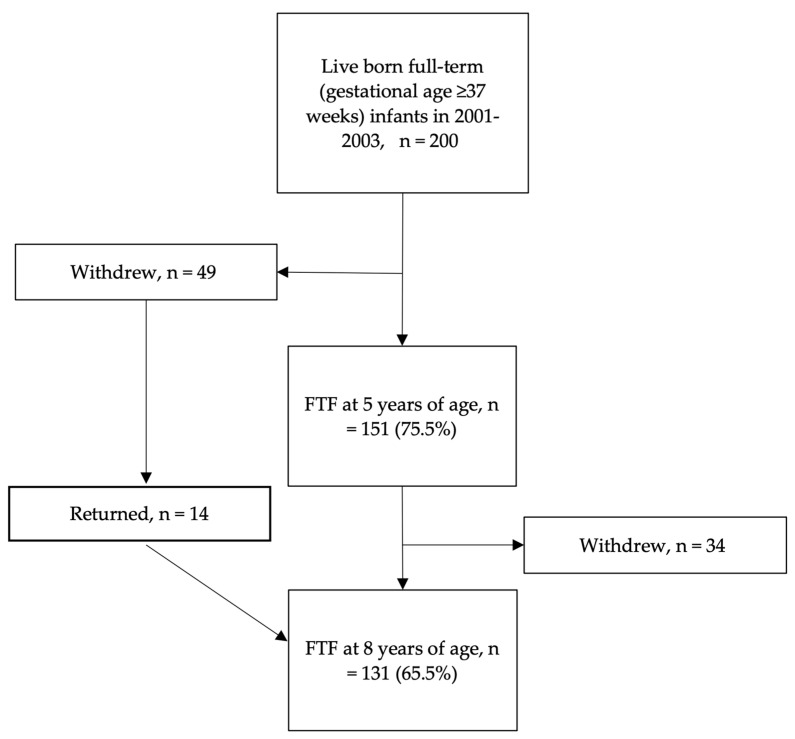
Flowchart of the full-term control group. FTF: Five-to-Fifteen questionnaire.

**Figure 3 jpm-13-00819-f003:**
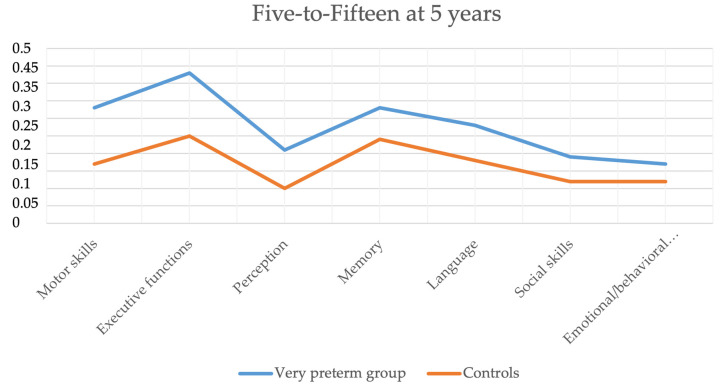
Five-to-fifteen profile of the children born very preterm and the control group at 5 years.

**Figure 4 jpm-13-00819-f004:**
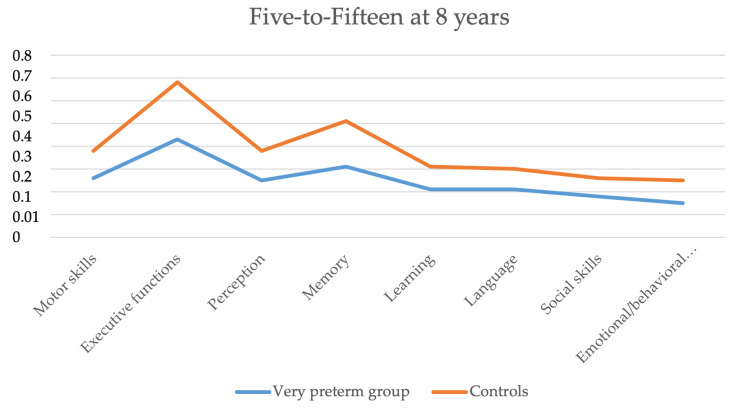
Five-to-Fifteen profile of the children born very preterm and the control group at 8 years.

**Table 1 jpm-13-00819-t001:** Background characteristics of the children born very preterm and full-term controls whose parents completed the Five-to-Fifteen questionnaire at the age of 5 and 8 years.

Background Characteristics	Children Born VeryPreterm		Controls	
	at 5 Years, n = 168	at 8 Years, n = 164	at 5 Years, n = 151	at 8 Years, n = 131
Gestational age, mean (SD), weeks	28.9 (2.8)	28.9 (2.8)		
Birth weight, mean (SD), grams	1104.7 (315.9)	1122.5 (323.8)	3652.7 (450.3)	3636.0 (437.4)
Birth weight z-score ^a^, mean (SD)	−1.4 (1.4)	−1.4 (1.5)		
Small for gestational age (<−2 SD), n (%)	55 (32.7)	53 (32.3)		
Prenatal corticosteroids, n (%)	161 (95.8)	154 (93.9)		
Multiple birth, n (%)	55 (32.7)	53 (32.3)		
Cesarean delivery, n (%)	107 (63.7)	98 (59.8)	23 (15.2)	21 (16.0)
Male, n (%)	93 (55.4)	77 (47.0)	69 (45.7)	61 (46.6)
Patent ductus arteriosus, operated, n (%)	22/167 (13.2)	17 (10.4)		
Bronchopulmonary dysplasia, n (%)	25 (14.9)	22 (13.4)		
Operated necrotizing enterocolitis, n (%)	9 (5.4)	8 (4.9)		
Sepsis, n (%)	35 (20.8)	31 (18.9)		
Laser-treated retinopathy of prematurity, n (%)	7/157 (4.5)	4/153 (2.6)		
Major brain pathology in MRI ^b^ at term, n (%)	38/166 (22.9)	40/160 (25.0)		
Mother’s education > 12 years, n (%)	76/158 (48.1)	72/155 (46.5)	50 (33.1)	46 (35.1)
Father’s education > 12 years, n (%)	42/158 (26.6)	39/155 (25.2)	43 (28.5)	37 (28.2)
Neurodevelopmental impairment, n (%)	12 (7.1)	14 (8.5)		
Cerebral palsy, n (%)	8 (4.8)	9 (5.5)		
Severe hearing impairment, n (%)	3 (1.8)	3 (1.8)		
Severe visual impairment, n (%)	0 (0)	0 (0)		
Severe cognitive impairment ^c^, n (%)	4/155 (2.6)	4/146 (2.7)		
Full-scale intelligence quotient, mean (SD)	102.1 (16.7)	101.9 (17.6)		

^a^ Birth weight in relation to gestational age. ^b^ The specific MRI protocol and details about the classification of the findings were previously described by Maunu et al. [24] and Setänen et al. [25]. ^c^ Full-scale intelligence quotient < 70 at 5 years; details about the classification were previously described by Lind et al. [26].

**Table 2 jpm-13-00819-t002:** The rate ratio (RR) estimates describe how many times higher the Five-to-Fifteen scores of 5-year-old children born very preterm were in comparison to the control group after adjustments for sex and the father’s educational level.

Five-to-Fifteen at 5 Years	Children Born Very Preterm, n = 168 Mean (Unadjusted)	Controls, n = 151 Mean (Unadjusted)	Adjusted RR	95% CI	*p*-Value
**Motor skills**	0.33	0.17	2.3 1.9 *	1.8–3.0 1.5–2.5	<0.001 <0.001
Gross motor skills	0.31	0.13	2.5 2.0 *	1.9–3.4 1.4–2.7	<0.001 <0.001
Fine motor skills	0.36	0.20	1.9 1.7 *	1.4–2.5 1.3–2.2	<0.001 <0.001
**Executive functions**	0.43	0.25	1.7 1.6 *	1.3–2.2 1.3–2.2	<0.001 <0.001
Attention and concentration	0.42	0.22	1.9 1.9 *	1.5–2.6 1.4–2.5	<0.001 <0.001
Overactivity and impulsivity	0.53	0.36	1.4 1.4 *	1.1–1.8 1.1–1.8	0.01 0.02
Passivity and inactivity	0.21	0.09	2.3 2.0 *	1.5–3.4 1.3–3.0	<0.001 0.001
Planning and organizing	0.41	0.25	1.5 1.4 *	1.1–2.1 1.0–2.0	0.02 0.04
**Perception**	0.21	0.10	1.9 1.8 *	1.4–2.5 1.3–2.3	<0.001 <0.001
Perception of space and direction	0.22	0.11	1.9 1.6 *	1.3–2.6 1.1–2.3	<0.001 0.007
Perception of own body	0.18	0.11	1.5 1.5 *	1.1–2.2 1.1–2.2	0.02 0.02
Perception of visual forms and figures	0.25	0.08	2.7 2.5 *	1.8–4.1 1.7–3.7	<0.001 <0.001
**Memory**	0.33	0.24	1.3 1.2 *	1.0–1.7 0.9–1.6	0.08 0.2
**Language**	0.28	0.18	1.5 1.3 *	1.1–1.9 1.0–1.7	0.003 0.03
Comprehension of spoken language	0.31	0.18	1.6 1.5 *	1.2–2.2 1.1–2.0	0.002 0.02
Expressive language	0.27	0.19	1.4 1.2 *	1.0–1.8 1.0–1.6	0.02 0.1
Verbal communication	0.31	0.17	1.8 1.6 *	1.3–2.6 1.1–2.4	0.001 0.009
**Social skills**	0.19	0.12	1.4 1.3 *	1.1–1.8 1.0–1.6	0.02 0.1
**Emotional/behavioral difficulties**	0.17	0.12	1.2 1.2 *	0.9–1.5 0.9–1.6	0.2 0.2
Internalization	0.13	0.09	1.5 1.4 *	1.1–2.0 1.0–1.9	0.02 0.02
Acting out	0.24	0.18	1.1 1.2 *	0.9–1.5 0.9–1.5	0.4 0.3
Obsessive actions or thoughts	0.09	0.05	1.6 1.6 *	1.1–2.4 1.1–2.3	0.01 0.03

* children with neurodevelopmental impairment excluded. The results of the comparisons are provided as rate ratios (RR) with 95% confidence intervals (CI) and *p*-values.

**Table 3 jpm-13-00819-t003:** The rate ratio (RR) estimates describe how many times higher the Five-to-Fifteen scores of 8-year-old children born very preterm were in comparison to the control group after adjustments for sex and the father’s educational level.

Five-to-Fifteen at 8 Years	Children Born Very Preterm, n = 164 Mean (Unadjusted)	Controls, n = 131 Mean (Unadjusted)	Adjusted RR	95% CI	*p*-Value
**Motor skills**	0.26	0.12	2.2 1.9 *	1.7–2.9 1.4–2.5	<0.001 <0.001
Gross motor skills	0.29	0.11	2.7 2.2 *	2.0–3.8 1.6–3.0	<0.001 <0.001
Fine motor skills	0.24	0.12	1.9 1.7 *	1.4–2.6 1.2–2.3	<0.001 0.001
**Executive functions**	0.43	0.25	1.5 1.5 *	1.2–2.0 1.1–1.9	0.001 0.005
Attention and concentration	0.52	0.30	1.7 1.6 *	1.3–2.2 1.2–2.1	<0.001 0.002
Overactivity and impulsivity	0.39	0.24	1.6 1.5 *	1.2–2.1 1.2–2.0	0.002 0.003
Passivity and inactivity	0.28	0.16	1.7 1.6 *	1.2–2.4 1.1–2.3	0.003 0.008
Planning and organizing	0.45	0.23	1.8 1.5 *	1.3–2.6 1.0–2.1	0.001 0.03
**Perception**	0.25	0.13	1.9 1.7 *	1.5–2.5 1.3–2.3	<0.001 <0.001
Perception of space and direction	0.19	0.07	2.7 2.5 *	1.8–4.0 1.6–3.8	<0.001 <0.001
Concepts of time	0.52	0.33	1.6 1.5 *	1.2–2.2 1.1–2.1	0.002 0.006
Perception of own body	0.18	0.10	1.8 1.5 *	1.2–2.6 1.0–2.3	0.004 0.04
Perception of visual forms and figures	0.13	0.03	4.1 3.1 *	2.3–7.3 1.7–5.7	<0.001 <0.001
**Memory**	0.31	0.20	1.5 1.5 *	1.2–2.0 1.1–2.0	0.002 0.007
**Learning**	0.21	0.10	1.9 1.8 *	1.4–2.6 1.3–2.5	<0.001 <0.001
Reading and writing	0.37	0.26	1.4 1.3 *	1.1–1.9 1.0–1.8	0.02 0.06
Arithmetic	0.35	0.21	2.0 1.9 *	1.4–2.8 1.3–2.7	<0.001 <0.001
General learning	0.24	0.10	2.3 2.2 *	1.6–3.4 1.4–3.2	<0.001 <0.001
Coping with learning	0.44	0.25	1.8 1.7 *	1.4–2.4 1.3–2.2	<0.001 <0.001
**Language**	0.21	0.09	2.2 2.1 *	1.7–2.9 1.6–2.8	<0.001 <0.001
Comprehension of spoken language	0.26	0.10	2.7 2.4 *	1.8–3.9 1.7–3.6	<0.001 <0.001
Expressive language	0.17	0.08	2.2 2.2 *	1.6–3.0 1.6–3.0	<0.001 <0.001
Verbal communication	0.27	0.14	1.8 1.8 *	1.2–2.7 1.2–2.7	0.002 0.005
**Social skills**	0.18	0.08	2.1 2.0 *	1.6–2.7 1.5–2.6	<0.001 <0.001
**Emotional/behavioral difficulties**	0.15	0.10	1.4 1.3 *	1.1–1.8 1.0–1.8	0.02 0.05
Internalization	0.14	0.09	1.6 1.5 *	1.1–2.1 1.1–2.1	0.005 0.008
Acting out	0.20	0.15	1.2 1.2 *	0.9–1.6 0.9–1.6	0.2 0.2
Obsessive actions or thoughts	0.07	0.05	1.5 1.4 *	1.0–2.2 0.9–2.1	0.08 0.1

* children with neurodevelopmental impairment excluded. The results of the comparisons are provided as rate ratios (RR) with 95% confidence intervals (CI) and *p*-values.

**Table 4 jpm-13-00819-t004:** Correlations between the FTF scores at 5 and 8 years in the children born very preterm and the controls.

FTF Domain	Children Born Very Preterm, n = 146, r (p)	Controls, n = 117, r (p)
**Motor skills**	0.76 (<0.001) 0.66 (<0.001), n = 136 *	0.70 (<0.001)
Gross motor skills	0.74 (<0.001) 0.62 (<0.001), n = 136 *	0.72 (<0.001), n = 116
Fine motor skills	0.65 (<0.001) 0.58 (<0.001), n = 136 *	0.58 (<0.001)
**Executive functions**	0.61 (<0.001), n = 145 0.60 (<0.001), n = 135 *	0.48 (<0.001)
Attention and concentration	0.58 (<0.001), n = 144 0.57 (<0.001), n = 134 *	0.46 (<0.001), n = 115
Overactivity and impulsivity	0.59 (<0.001), n = 145 0.57 (<0.001), n = 135 *	0.44 (<0.001)
Passivity and inactivity	0.48 (<0.001) 0.48 (<0.001), n = 136 *	0.35 (<0.001)
Planning and organizing	0.49 (<0.001), n = 144 0.48 (<0.001), n = 134 *	0.42 (<0.001), n = 116
**Perception**	0.56 (<0.001), n = 144 0.50 (<0.001), n = 135 *	0.31 (0.001), n = 116
Perception of space and direction	0.39 (<0.001) 0.30 (<0.001), n = 136 *	0.47 (<0.001)
Perception of own body	0.55 (<0.001), n = 144 0.57 (<0.001), n = 135 *	0.37 (<0.001), n = 116
Perception of visual forms and figures	0.57 (<0.001), n = 144 0.52 (<0.001), n = 135 *	0.19 (0.04), n = 116
**Memory**	0.61 (<0.001), n = 144 0.62 (<0.001), n = 135 *	0.62 (<0.001), n = 116
**Language**	0.75 (<0.001) 0.77 (<0.001), n = 136 *	0.44 (<0.001), n = 115
Comprehension of spoken language	0.44 (<0.001), n = 145 0.45 (<0.001), n = 135 *	0.27 (0.003), n = 115
Expressive language	0.77 (<0.001) 0.80 (<0.001), n = 136 *	0.31 (0.001), n = 115
Verbal communication	0.56 (<0.001) 0.56 (<0.001), n = 136 *	0.59 (<0.001), n = 115
**Social skills**	0.72 (<0.001) 0.70 (<0.001), n = 136 *	0.52 (<0.001), n = 116
**Emotional/behavioral difficulties**	0.66 (<0.001), n = 145 0.67 (<0.001), n = 135 *	0.69 (<0.001)
Internalization	0.52 (<0.001), n = 145 0.52 (<0.001), n = 135 *	0.71 (<0.001)
Acting out	0.70 (<0.001), n = 145 0.70 (<0.001), n = 135 *	0.68 (<0.001)
Obsessive actions or thoughts	0.54 (<0.001), n = 145 0.59 (<0.001), n = 135 *	0.31 (0.001), n = 116

* children with neurodevelopmental impairment excluded. Correlations were calculated using Pearson’s rho. *p* < 0.05 was considered significant.

## Data Availability

The data presented in this study are available on request from the corresponding author. The data are not publicly available due to privacy of the children involved in the study.

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
