# Peer review of "Five-to-Fifteen—Parental Perception of Developmental Profile from Age 5 to 8 Years in Children Born Very Preterm"

_jpm, 2023, doi:10.3390/jpm13050819_

Round 1

Reviewer 1 Report

The study aimed to support the use of a parental questionnaire in the assessment and early detection of impairment i preterm children at school age. This could have clinical relevance given the cost of specialistic assessment. Nevertheless, some relevant methodological issue should be observed. Mainly, I did not agree that the study could give an overview of the developmental profile given that an assessment by professional clinical is missing. In absence of it, authors could only report which is the parental perception of their children. despite this result is very interesting and have clinical implication (parents interacts with children according of their representation), we can't consider it as an objective assessment. for my knowledgment (and according to what authors declared) no study assessed the convergent validity  between FTF and one developmental scale. so, no theoreotical is given to support the use of term as developmental profile. I suggest to author to add this info (if it exist) or to adjust using the term parental perception of developmental profile.

Another methodological concerns is about the sample recruited. In flow chart I understand that from original sample 2 group of preterm children agreed to partecipate to the follow up study. Nevertheless, authors did not explain how many children are the same in the two group. Given that authors compared these two group and consider the correlation of two scores, these value should be comparable (i.e from the same group). Moreover, as authors briefly suggested in limit, they are not sure that the same parents fulfilled the questionnaire. I think that limit the study at the description of the results at two age without any comparison could be more adequate considering all these criticity.

I also suggest to include in Introduction more literature in the advantage of the use of parental questionnaire in the assessment of children development, in order to clarify the dubts emerged.

Another consideration regard the description of sample. Descriptive variables are given for preterm children group compared with withdrawn ones. nevertheless, no comparison between control group is given. Are they significantly different?

in the results section I did not understand why n the text some results are described and some other not. Please level out them.

the tables are not formatted according to journal. specifically, put the mean of * in the note and not in title.

Figure 2 and 3 are not cited in the text. so I did not understand what they mean.

finally, the discussion did not consider all the wide literature on development of preterm children. Authors cited only results for some scale and not for all. Specifically, a possible explanation of the few not significant scales should be given.

Reviewer 2 Report

Reviewer Comment

Although the study is a valuable study in terms of evaluating and examining developmental areas in different age categories in preterm infants, the importance and justification of the study should be better expressed. Comparisons and statements were made on the questionnaire in many parts of the Manuscip. In this way, development parameters should be emphasized instead of expression. Your comparison here should not be a survey. Motor, behavior, cognitive. etc headers. Although you compare this in all your analyses, more emphasis is placed on the survey, especially the title. This form of expression should change. In addition, there are places in the manuscript that do not comply with the article writing system. The purpose and rationale of the study should be better expressed. It is wrong to clearly state that there is no such study. You should give justification considering the possibility of not being able to reach such a study in your literature search. The method section should be expressed more systematically. Presenting the participants, study design, evaluation measures, and statistical analysis sections under the headings will make the manuscript appear more compact. In addition, there are some missing information under these headings, and this information should be added as stated below. Statistical analysis should be stated under the table in the conclusion section. and individuals should be presented with demographic data within both groups. The discussion section was the weakest part of the manuscript. The research results were not sufficiently discussed with the literature and the conclusions made included too many personal comments. If you have given the results of all development parameters, each of these results should be analyzed in detail with the literature and inferences should be made by showing the literature source. When the sources are examined, it is seen that some sources are old. Replacing these sources with newer ones will improve the quality of the manuscript. With these proposed regulations, the manuscript will reach a better level. Please make all arrangements carefully and neatly.

Revision

1.      Page 1, line 1-3; “Five-to-Fifteen developmental profile at age 5 and 8 years in children born very preterm” The title of the study is considered insufficient in terms of reflecting the content of the research. You used a very general expression in the title of the study, it is not clear what you are examining about what was done (eg comparison, intervention, prevalence, etc.). Edit the title to express the target point of your research.

2.      Page 1, line 27; Keywords should be updated. You made developmental assessment as the main emphasis in your study, but did not specify it in these keywords. Edit the keywords to highlight this.

3.      Page 1, line 40,41; “Developmental assessments are numerous, often time consuming and requiring licensed education as well as resources” After this statement, you mentioned previous studies. Since you are switching to another sentence and explanation that does not have a continuation with this sentence, when there is a different transition in terms of both meaning and subject, reference must be made. You should also include more than one source because you cite more than one evaluation questionnaire. Include a reference of the source from which you obtained this information.

4.      Page 2, line 48-51; “There are no prospective studies comparing the whole FTF profile of children born very preterm and controls at different age points with the aim of discovering correlations between the FTF scores at different age points.” While explaining your purpose and reason for doing the study, it is not appropriate to use a clear and precise statement as there is no such study. You should state this as you could not find such a study in the literature reviews. You should also take into account the possibility that you may not be able to find it yourself. You should also better articulate the rationale for the study. You should explain why the work you do is important, why it is important to do it in a different age category, and what it will lead to learning. Make the necessary arrangements.

5.       Page 2, line 57; The material and method section should be expressed more systematically. This section should consist of 4 subsections. In this way, the design of the study will be better and systematically read and understood.

1. Study Design: Here you should describe the overall design of the study. First of all, the short plan of the study should be stated. A reader reading this will easily understand the subject of the study. It should be explained who the groups are and what evaluations will be made. Ethics committee permission (number and date) and information about where and between what dates the study was conducted should be shared here. It should also be noted that the assessments are made by experts with many years of experience in this field.

2. Participant: The inclusion and exclusion criteria of individuals in both groups should be explained here. The age range of these groups should be specified. It should be explained how the number of individuals is determined. Information should be given about whether the power analysis was done, how it was done, and how many people were included in the study as a result. Flow chart of individuals should be given in this section.

3. Evaluations: Information about the evaluation questionnaire should be disclosed here. It should be explained who created the questionnaire used, and what the reliability coefficient is. It should also be explained how many years of experience the evaluations are made by experts in this field.

4. Data analysis: After explaining which program the analysis will be carried out, it should be stated how the normality of the data is. It should be stated whether the data are normally distributed and which statistical analysis method will be used accordingly. Then, it should be explained which statistical method will be used for which analysis.

6.      More information should be given about the children in the control group. These children should be included in the flow chart of individuals. In addition, the children in the control group should also be given background characters as in table 1. The importance of giving this information is to present and know the norm values.

7.      Make a statistical comparison of the data belonging to the two age categories in Table 1. Present in Table 1 by expressing it as a column with p significance value.

8.      “The difference  between the groups was not statistically significant in Memory, Emotional/behavioral difficulties and Acting out. When the children with neurodevelopmental impairment were excluded, the difference between the groups was no longer statistically significant in Expressive language and Social skills (Table 2).” Add the p significance values.

9.      Include the name of the statistical analysis method used under all tables. Also, express the significance value of p under the table and highlight the p values that are significant in the table.

10.  “The difference between the groups was not statistically significant in Acting out and Obsessive actions and thoughts. When the children with neurodevelopmental impairment were excluded, the difference between the groups was no longer statistically significant in Reading and Writing and Emotional/behavioral difficulties (Table 3).”  In all texts, also express the significance values of p where you say statistically significant.

11.  Page 12, 195; line The discussion is written very superficially. All subheadings need to be discussed. While discussing, more relations should be established with the studies in the literature. There are many personal comments in the comments and inferences made. and inferences should be made with the support of the literature. The discussion should be seriously regulated in this sense. The discussion should be based on your own data and an in-depth review of the literature. The discussion should be organized in detail with these suggestions.

12.  Page 16, line 316; You should use up-to-date and new sources as much as possible in the sources you use in manuscript writing. It seems that some sources are not old. Revise these resources with more up-to-date and new ones.

Round 2

Reviewer 1 Report

I thank authors for their work that improved the quality of manuscript. 

Nevertheless, I had two doubts about the revision. First, in their comments authors declared that only parents that fulfilled questionnaire at both age will be included in the study. nevrtheless at row 76 they wrote "All children whose parent had completed the FTF at 5 and/or at 8 years were 76 included. No power analysis was performed". again, in analyses section they did not explain that correlation regard only a part of the sample. Please, clarify this aspect.

second, in discussion authors added comments on results on Memory and Emotional/behavioral difficulties scores, as sugegsted. I further suggest to enforce the disccussion, given more a description on the change in the time (memory scores are signficant at 8 years, emotional at 5 years). also a comment on comparison with studies that assessed these area by clinicans perspective could be useful. Could be possible that parents are less able to assess these abilities rather than other ones (i.e. language, motor, etc.)? I think that this consideration could deep the strenght of results and open to new research question. 

Reviewer 2 Report

Reviewer Comment

Although most of the revisions suggested in the review of the study have been made, there are still areas that need to be corrected. These places are listed below. By correcting these suggestions, the manuscript will reach a better level. Please make all additions completely.

Revision

1.      Page 2, line 52-54;“To the best of our knowledge, there are no prospective studies comparing the whole FTF profile of children born very preterm and controls at different age points with the aim of discovering correlations between the FTF scores at different age points. Instead of saying it with a clear statement like you don't have a statement here. You should better evaluate the possibility of not being able to reach your studies. This statement should be in the form that we could not find such a study in our scans. Make the necessary correction.

2.      Page 2, line 55-58; It is important to study the parental perception of developmental difficulties and their persistance into school age in children born very preterm. With the help of this information we could provide early identification of developmental problems and allocate resources for timely support and prevention.” There are informative expressions in these sentences and you should definitely cite the source for them.

3.      Page 2, line 67; In the study design section, add information about how the number of participants was determined. Has power analysis been done? If so, how was the Power analysis done, with reference to what? Add their information.

4.      Page 6, line 155; “The difference in continuous background characteristics between the children born very preterm participating the study and those who withdrew were analyzed using either the two-sample t-test when variables were normally distributed or the Mann-Whitney U Test when variables were not normally distributed. For the categorical background characteristics, a chi-square test or Fisher’s exact test was used.” Information on how to test for normality should be added to the beginning of this paragraph.

5.      Page 9, line 219; Information on which statistical analysis was performed should be added under Table 2.

6.      Page 12, line 247; Information on which statistical analysis was performed should be added under Table 2.
